# Photodynamic versus white light-guided treatment of non-muscle invasive bladder cancer: a study protocol for a randomised trial of clinical and cost-effectiveness

Zafer Tandogdu,[1] Rebecca Lewis,[2] Anne Duncan,[3] Steven Penegar,[4] Alison McDonald,[3] Luke Vale,[5] Jing Shen,[6] John D Kelly,[7] Robert Pickard,[8] James N Dow,[9] Craig Ramsay,[10] Hugh Mostafid,[11] Paramananthan Mariappan,[12] Ghulam Nabi,[13] Joanne Creswell,[14] Henry Lazarowicz,[15] John McGrath,[16] Ernest Taylor,[17] Emma Clark,[18] Graeme Maclennan,[3] John Norrie,[19] Emma Hall,[20] Rakesh Heer,[1] PHOTO Trial Management Group

ZT, RL, AD and SP contributed equally.

For numbered affiliations see end of article.

**Correspondence to**
Rakesh Heer;
rakesh.heer@newcastle.ac.uk

## ABSTRACT

**Introduction** Bladder cancer is the most frequently occurring tumour of the urinary system. Ta, T1 tumours and carcinoma in situ (CIS) are grouped as non-muscle invasive bladder cancer (NMIBC), which can be effectively treated by transurethral resection of bladder tumour (TURBT). There are limitations to the visualisation of tumours with conventional TURBT using white light illumination within the bladder. Incomplete resections occur from the failure to identify satellite lesions or the full extent of the tumour leading to recurrence and potential risk of disease progression. To improve complete resection, photodynamic diagnosis (PDD) has been proposed as a method that can enhance tumour detection and guide resection. The objective of the current research is to determine whether PDD-guided TURBT is better than conventional white light surgery and whether it is cost-effective.

**Methods and analysis** PHOTO is a pragmatic multicentre randomised controlled trial (open parallel group, non-masked and superiority trial) comparing the intervention of PDD-guided TURBT with standard white light resection in newly diagnosed intermediate and high risk NMIBC within the UK National Health Service setting. Clinical effectiveness is measured with time to recurrence. Cost-effectiveness is assessed within trial via the calculation of incremental cost per recurrence avoided and incremental cost per quality-adjusted life per year gained over 3 years and over long term through a modelling exercise over patients' lifetime.

**Ethics and dissemination** Formal ethics review was undertaken with a favourable opinion, in line with UK regulatory procedures (REC reference number: 14/NE/1062). If reductions in time to recurrence is associated with long-term patient benefits, the cost-effectiveness evaluation will provide further evidence to inform adoption of the technology. Findings will be shared in lay media such as patient and charity forums and will be presented at key meetings and published in academic literature. Trial registration number ISRCTN84013636.

## Strengths and limitations of this study

► The effectiveness of the photodynamic diagnosis (PDD) for the initial resection of intermediate and high risk non-muscle invasive bladder cancer (NMIBC) as part of routine care will be demonstrated with a pragmatic clinical trial design.
► Full-health economic evaluation will provide high-quality evidence of the burden of NMIBC for the National Health Service.
► A well-characterised trial associated biorepository of longitudinal serially collected tissue samples.

## BACKGROUND

Bladder cancer is the most frequently occurring tumour of the urinary system.[1] Staging of bladder cancer is described using the tumour, node and metastasis system.[2] Tumours confined to the epithelial lining (urothelium) are classified as stage Ta, and those invading the lamina propria are classified as stage T1.

Ta and T1 tumours can be removed by transurethral resection (TURBT) that involves passing a cystoscope through the urethra into the bladder and resecting the tumour under direct visualisation. For therapeutic purposes, Ta and T1 tumours are grouped together as non-muscle invasive bladder cancer (NMIBC). Grade (microscopic characteristics of the tumour cells) can be used to describe aggressiveness of cancers and are characterised as either low grade (relatively benign) or high grade (aggressive). NMIBC also include flat, high-grade tumours that are

BMJ

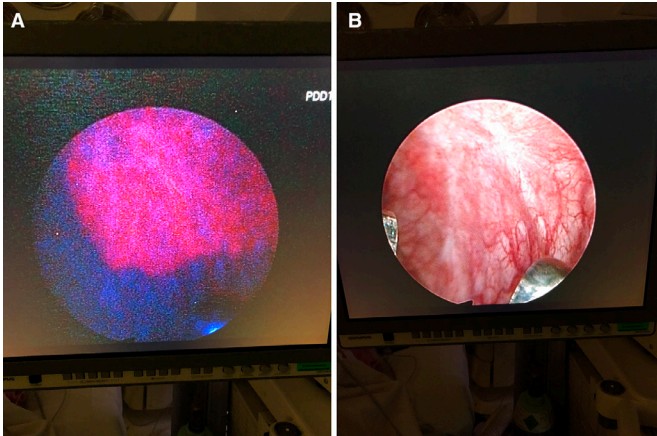

**Figure 1** White light (A) and blue light (photodynamic) (B) cystoscopy image of the bladder from a patient diagnosed with CIS. On PDD, the area with CIS appears red, while with WL the area is unclear. CIS, carcinoma in situ; PDD, photodynamic diagnosis.

confined to the epithelium classified as carcinoma in situ (CIS).[2]

Complete resection of the tumour with TURBT is essential to obtain good prognosis. It is thought that failure to identify satellite tumours or to appreciate the full extent of the tumours visualised during resection using conventional white light cystoscopy may be a factor in 20%–40% of recurrent bladder tumours.[3 4] Incomplete resection with TURBT is also associated with staging errors. In order to correct the staging errors associated with initial TURBT, a second resection within 2–6 weeks is suggested for select group patients.[5] It has been postulated that development in cystoscopy imaging can improve resections and decrease the need for a second resection.[6]

Recurrence and stage progression to muscle invasive (T2–T4) or metastatic cancer is more likely to occur in those with high-grade tumours with concomitant CIS. CIS in particular can be easily missed using conventional white light-guided resection.[6]

Surveillance of NMIBC is carried out with cystoscopy to detect recurrence early and allow treatment before progression. Clinical guidelines tailor follow-up protocols according to the risk groups (low, intermediate and high) developed using clinical and histological parameters.[7] Advised follow-up of low risk is at 3 months, and if negative, the next cystoscopy is scheduled for 9 months later and then yearly for 5 years. Patients with high-risk tumours have cystoscopy and urine cytology at 3 months. If negative, it is repeated every 3 months for 2 years, then every 6 months until 5 years and annually thereafter.[5] The intensity of cystoscopic follow-up for patients with intermediate risk is not clearly defined, for which a follow-up scheme in-between those described for low and high risk and is adapted according to personal and subjective factors.[5]

## Photodynamic diagnosis (PDD) of NMIBC

As an attempt to improve resection rates, PDD has been developed to enhance tumour detection and guide resection. A cystoscopy image of white light (WL) versus PDD is presented in figure 1. Meta-analyses and systematic reviews of PDD-guided treatment of NMIBC have shown efficacy in tumour detection and reduction in residual tumour compared with white light cystoscopy alone. These findings translate into reduced recurrence rates.[6 8] However, these trials were efficacy studies, and the systematic review called for a pragmatic study to allow better interpretation of possible benefit into daily clinical practice.

## Health economics of NMIBC

NMIBC is one of the most costly cancers to manage on a per-patient basis because of its high prevalence, high recurrence rate, need for adjuvant treatments and the requirement for long-term cystoscopic surveillance. The total cost of treatment and 5-year follow-up of patients with NMIBC diagnosed in the UK has increased from £73 million to £213 million from 2001 to 2012 (inflation corrected).[9 10] From a patient perspective, there often are considerable anxieties about recurrences, transurethral resection and progression, requiring additional therapies with potential mortality and long-term morbidity (eg, radical surgery). Transurethral resection itself is associated with reduced quality of life, including both mental and physical health domains, although these effects are usually transient.[11] Substantial effects on health-related quality of life (HRQoL) are most likely to come from adjuvant intravesical treatments and radical or palliative treatments for progression.[12] The cost-effectiveness of NMIBC treatment strategies has not been widely studied.

## NMIBC biomarkers and clinical impact

To date, existing non-invasive commercial biomarkers (primarily urinary) are not embedded in routine clinical practice due to poor sensitivity, specificity and lack of evidence. Several research bodies have recognised the lack of clinically useful biomarkers for bladder cancer. 'Fit for purpose' sample resources accessible to high-throughput 'omic' technologies will afford the greatest opportunity to generate translational hypotheses and ensure clinical validity and utility of putative candidate markers/signatures.[13 14] Robust, 'future-proof', longitudinal serial sample archives providing critical insights of the natural history of bladder cancer correlated with clinical detail for retrospective translational biomarker discovery are lacking.

## Current research objectives

More efficient management strategies to reduce NMIBC recurrence and hence decrease both the burden to patients and costs are urgently needed. PDD-guided initial TURBT has been identified as a technique that can help achieve these aims. The objective of the current research (PHOTO trial) is to determine whether photodynamic surgery guided by a fluorescent tumour marker is better

than conventional white light surgery in the cystoscopic treatment of people with intermediate and high risk cancers confined to the bladder lining and whether its implementation is cost-effective. The trial includes a full assessment of the costs of patient management through the care pathway. Individual patient data from this trial will be used for subsequent mathematical modelling studies to investigate safe monitoring frequency. The photodynamic versus white light-guided treatment of NIMBC trial has the following research objectives:

### Primary objectives

Clinical effectiveness: to compare time to recurrence for each of the two treatment strategies, with a principal point of interest at 3 years.

Cost-effectiveness: to evaluate cost-effectiveness as measured by the incremental cost per recurrence avoided and the cost-utility as measured by the incremental cost per quality-adjusted life year (QALY) gained at 3 years and over patients' lifetime.

### Secondary objectives

a. To measure relative rates of disease progression at 3 years.
b. To measure relative harms and safety.
c. Patient lifetime HRQoL and cancer-specific survival.

### Additional objectives

a. To model the safest and most cost-effective cystoscopic follow-up surveillance schedule.
b. To evaluate the learning curve for the procedure and account for its effects on outcomes of both PDD-guided and standard white light resections.
c. To establish a well-characterised cohort of serial samples from patients with intermediate and high-risk NMIBC including clinical data, urine, blood and tumour specimens for separately funded translational research.

## METHODS AND DESIGN
### Study design

PHOTO is a multicentre randomised open parallel group non-masked superiority trial comparing the intervention of PDD guided bladder tumour resection with standard white light resection in patients with newly diagnosed intermediate and high risk NMIBC. Apart from initial treatment, both groups will receive standard care, including single dose intravesical mitomycin C within 24 hours of initial resection, surveillance according to risk-adjusted schedules and adjuvant therapy as indicated by current practice guidelines. The target number of patients to be recruited is 533 with a trial specific follow-up of at least 36 months for each individual. The outline of the study protocol (version 4, 2 February 2018) is shown in figure 2.

### Intervention

The interventions being compared within PHOTO trial are: PDD-guided transurethral resection (TURBT) (experimental group) versus standard white light TURBT (control group).

All participants, unless there are clinical contra-indications, receive intravesical mitomycin C (40 mg in 40 mL saline), ideally within 6 hours following TURBT but otherwise within the inpatient setting before discharge.

### Technique of PDD

PDD requires preliminary instillation of the photosensitiser hexaminolevulinate (85 mg in 50 mL of phosphate buffered saline) into the participant's bladder through a urethral catheter. Participants are asked not to pass urine for at least 1 hour after insertion. Following operating theatre preparation according to local standard procedures and under appropriate anaesthesia, participants undergo TURBT of their bladder tumour under blue light (wavelength 380–450 nm) illumination of the bladder. The equipment required includes a specialised light source, cystoscope, light cables and cameras. When using PDD, normal bladder epithelium appears blue, while red areas should be considered suspicious and should be resected.

### Technique of standard white light cystoscopy

The control group does not have any preliminary photosensitiser instillation and undergo standard tumour localisation and resection under white light (wavelength 400–800 nm) illumination of the bladder.

### Participants

In the PHOTO trial, adult patients with a suspected new diagnosis of intermediate or high risk NMIBC are studied. Participants are identified prior to initial resection based on results of preliminary visual assessment via cystoscopy or imaging performed as part of standard evaluation for suspected urinary tract malignancy.

Patients with the following criteria are included in the PHOTO trial:

► Adult men and women aged ≥16 years.
► First suspected diagnosis of bladder cancer.
► Visual/ultrasound/CT diagnosis of intermediate/high risk NMIBC.
► White light visual appearances of intermediate or high risk disease (≥3 cm or two or more tumours or flat velvety erythematous changes alerting a clinical suspicion of CIS).
   OR
   Suspicion of papillary bladder tumour ≥3 cm based on ultrasound or CT scanning (without hydronephrosis).
► Written informed consent for participation prior to any study specific procedures.
► Willing to comply with the following lifestyle guidelines:
   – Female participants must be surgically sterile or be postmenopausal or must agree to use effective contraception after joining the study and for 7 days after treatment. Female participants must not breast feed for 7 days after treatment.

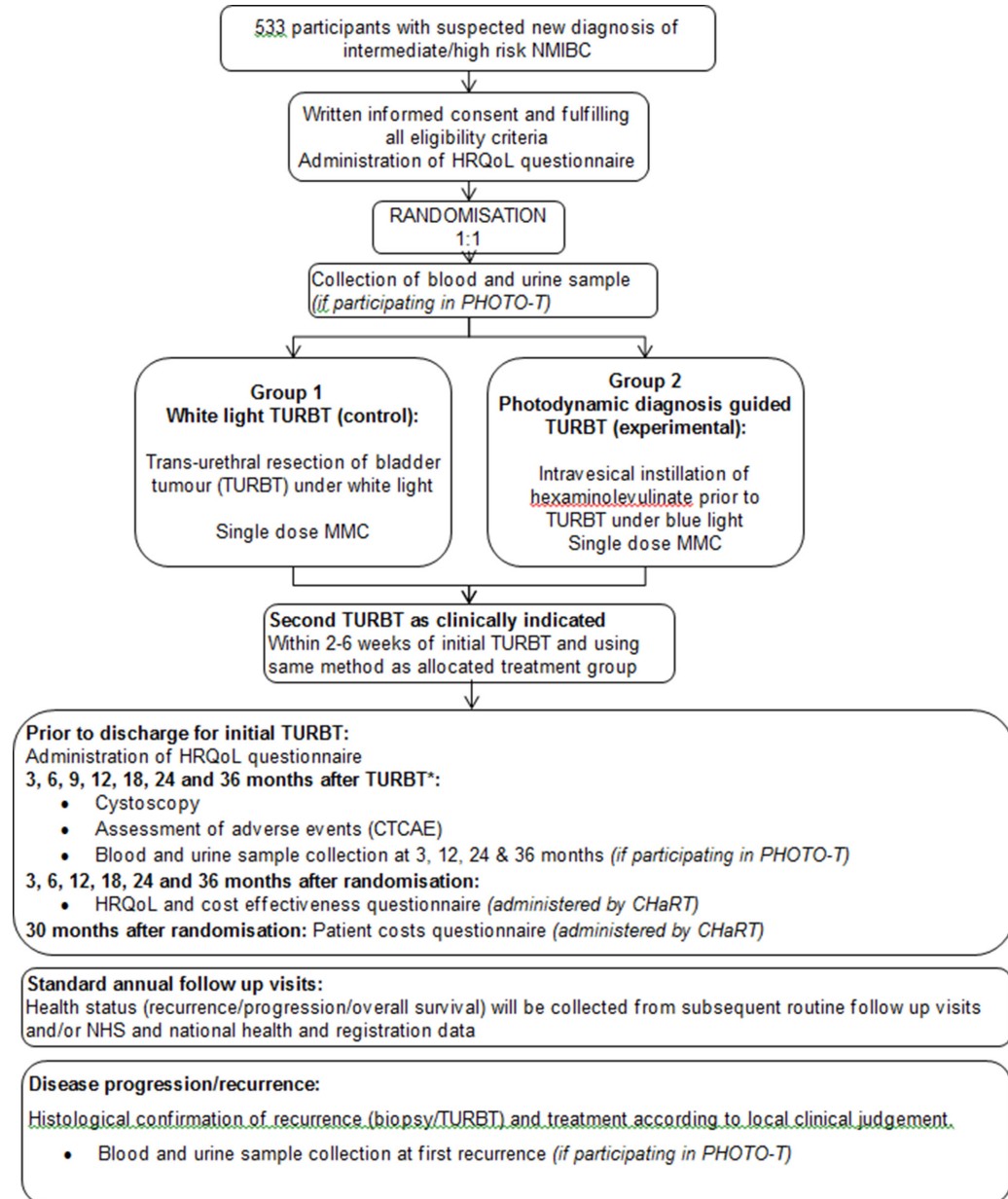

**Figure 2** PHOTO trial study design summary.

– Male participants must be surgically sterile or must agree to use effective contraception after joining the study and for 7 days after treatment.

– Effective contraception is defined as two forms of contraception, including one barrier method.

Exclusion criteria applied in the PHOTO trial are:

► Visual evidence of low risk NMIBC (solitary tumour <3 cm).

► Visual evidence of MIBC on preliminary cystoscopy, that is, non-papillary or sessile mass (attached directly by its base without a stalk).

► Imaging evidence of MIBC – CT/USS (this includes the presence of hydronephrosis, which may be present despite clear imaging of MIBC in the bladder).

► Upper tract (kidney or ureteric) tumours on imaging.

► Any other malignancy in the past 2 years (except: non-melanomatous skin cancer cured by excision, adequately treated CIS of the cervix, Ductal carcinoma in situ (DCIS) / Lobular carcinoma in situ (LCIS) of the breast or prostate cancer in patients who have a life expectancy of >5 years on trial entry).

► Evidence of metastases.

► Porphyria or known hypersensitivity to porphyrins.

► Known pregnancy (based on history and without formal testing, in keeping with day-to-day National Health Service (NHS) practice of PDD use).

► Any other conditions that in the principal investigator's opinion would contraindicate protocol treatment.

► Unable to provide informed consent.

► Unable or unwilling to complete follow-up schedule (including HRQoL questionnaires).

## Informed consent-ethics approval

The study complies with the Helsinki Declaration and the principles of Good Clinical Practice.

Potential participants are identified mainly through rapid access haematuria clinics at participating sites from the UK. An eligibility checklist is completed by the local principal investigator (or delegate) to assess fulfilment of the entry criteria for all patients considered for the study. Information from the diagnostic cystoscopy is used to assess eligibility.

All potentially eligible patients are provided with an information sheet to explain why they have been approached and the nature of the study. Eligible patients are asked to provide written informed consent for the study only after they have had sufficient time to consider the trial and had the opportunity to have any further questions addressed.

## Recruitment and randomisation

Eligible patients are centrally randomised using either the secure web-based or the 24-hour interactive voice response randomisation system at the Centre for Healthcare Randomised Trials in Aberdeen, using minimisation by centre and gender, to allocate participants 1:1 to the control and experimental groups. The minimisation algorithm incorporates a random element in order to prevent deterministic treatment allocation. The trial was opened on the 23 October 2014 and recruitment was completed on 14 February 2018. Over this period, 22 centres joined the study and contributed to recruitment. List of sites participating are available on http://www.isrctn.com/ISRCTN84013636.

## Outcome measures

### Primary outcome measures

Clinical effectiveness: time to recurrence is measured as time from randomisation to first recurrence.

Cost effectiveness: a health economic model will be developed to calculate incremental cost per recurrence avoided and incremental cost per QALY gained over 3 years.

### Secondary outcome measures

Clinical effectiveness:
► Adverse events and complications up to 3 months from initial or second TURBT are captured and will be included in analysis.
► HRQOL is captured for each participant at baseline (prior to knowledge of treatment allocation), following surgery and at 3, 6, 12, 18, 24 and 36 months after randomisation.
► Disease progression is captured within the trial time. Patient lifetime projections will be made by using the trial data at baseline and supplementing as necessary from other published data.

► Overall survival and bladder cancer-specific survival will be compared between the two treatment arms. Minimum follow-up of the last included patient will be 3 years and maximum expected follow-up is 66 months.

Cost effectiveness:
► Estimation of the incremental cost per recurrence avoided using the economic model over the patients' lifetime.
► Estimation of the incremental cost per QALY gained using the economic model over the patients' lifetime.

### Additional outcomes measures

Schedules for follow-up: using data from within the trial and, if appropriate, from other relevant sources, the risk of recurrence at each interval surveillance cystoscopy will be described to then model the most safe and efficient surveillance follow-up schedule.

The effect of PDD guided resection experience (learning curve) on clinical effectiveness: a subgroup analysis comparing outcomes from PDD-experienced and PDD-naïve surgeons (determined at baseline) will be conducted. Also for PDD-naïve surgeons, an assessment of learning curve will be undertaken by comparing increasing experience and recurrence, in both PDD and WL resections.

## Tracking and monitoring adverse events

Direct surgically related postoperative events occurring within 30 days following the first TURBT or second TURBT if required will be assessed using The Clavien Dindo classification for surgical complications. Events occurring up to 3 months after TURBT (second TURBT if required) will be assessed and recorded using the National Cancer Institute Common Terminology Criteria for Adverse Events V.4.0 framework (http://ctep.cancer.gov/).[15]

## Trial assessment and measures

PHOTO trial schedule of assessment and investigations are summarised in table 1. Routine attendances for diagnosis and staging of new bladder cancers are used to establish eligibility, which includes obtaining the medical history. Eligible patients who consent for the trial are administered HRQoL questionnaires prior to primary TURBT and prior to discharge.

Time to recurrence will be measured from the day of randomisation to the day of subsequent biopsy with pathologically proven recurrence. Data will be collected from the following routine visits; 3, 6, 9, 12, 18, 24 and 36 months postinitial TURBT (or second TURBT if required). Associated costs and changes in HRQoL will be measured. These will be collected by postal questionnaires sent directly to participants at 3, 6, 12, 18, 24 and 36 months postrandomisation.

Disease progression will be assessed using results of further resection or imaging during follow-up. Progression will be defined as increase of stage into MIBC or

**Table 1** Schedule of investigations/assessments in PHOTO trial

| Visit/assessment | Pre-randomisation screening | Pretreatment TURBT | Prior to discharge | Second TURBT (as clinically indicated) | 3 months post-treatment | 6 months post-treatment | 9 months post-treatment | 12 months post-treatment | 18 months post-treatment | 24 months post-treatment | 36 months post-treatment | Annually thereafter | At first disease recurrence /progression |
|---|---|---|---|---|---|---|---|---|---|---|---|---|---|
| | | | | **Surveillance** | | | | | | | | | |
| Visual diagnosis of IR/HR NMIBC | X | | | | | | | | | | | According to EAU guidelines | Treatment according to local practice |
| Medical history | X | | | | | | | | | | | | |
| HRQoL questionnaire* | X | | X | X | X† | X† | X† | X† | X† | X† | X† | | |
| TURBT according to treatment allocation with post-treatment MMC instillation | | X | | | | | | | | | | | |
| Second TURBT, if required, according to treatment allocation | | | | X | | | | | | | | | |
| Assessment of adverse events (CTCAE and Clavien Dindo) | | | | | X | | | | | | | | |
| Cystoscopy | | | | | X | X | X | X | X | X | X | | X |
| Histological confirmation of recurrence/progression | | | | | | | | | | | | | X |
| Collection of FFPE tissue‡ | | X | | | | | | | | | | | X |
| Urine sample collection‡ | X | | | | X | | | X | | X | X | | X |
| Blood sample collection‡ | X | | | | X | | | X | | X | X | | X |

*European Organization for Research and Treatment of Cancer Quality of Life Questionnaire Version 3.0 (EORTC QLQ-C30), European Organisation for Research and Treatment of Cancer (EORTC) Quality of Life Questionnaire for Non-Muscle-Invasive Bladder Cancer (QLQ-NMIBC24) (NMIBC24) and EQ-5D-3L.

†Questionnaire sent by post directly to participant.

‡If patient consented to participation in PHOTO-T (as this is archived pathology the tissue may be requested at an interval from the diagnostic resection/recurrence).

CTCAE, Common Terminology Criteria for Adverse Events; FFPE, Formalin-Fixed, Paraffin-Embedded; HR, High risk; HRQoL, health-related quality of life; IR, Intermediate risk; MMC, Mitomycin-C; NMIBC, non-muscle invasive bladder cancer; PHOTO-T, PHOTO Translational; TURBT, Transurethral resection of bladder tumor.

development of nodal or metastatic disease. In addition, patients showing failure to respond to intravesical treatment (eg, BCG failure) will also be captured.

The relative changes in HRQoL resulting from the physical and psychological benefit together with any harms associated with each strategy and with subsequent necessary cancer treatment will be measured using the generic EQ-5D-3L questionnaire, cancer-specific European Organization for Research and Treatment of Cancer Quality of Life Questionnaire Version 3.0 (EORTC-QLQ-C30) and disease-specific European Organisation for Research and Treatment of Cancer (EORTC) Quality of Life Questionnaire for Non-Muscle-Invasive Bladder Cancer (EORTC QLQ-NMIBC24) questionnaire completed by the participant.

Effect of PDD-guided resection experience on clinical effectiveness: all recruiting surgeons will complete a learning curve questionnaire to elicit their white light and PDD resection experience prior to any recruitment. The subsequent accruing experience of each surgeon will be captured on case report forms. Early recurrence (12 weeks) will be used as a proxy of incomplete resection.

## Sample size

We aim to detect an absolute reduction in recurrence at 3 years of 12%: from 40% (under the conservative assumption that all the patients recruited are intermediate risk patients with a probability of recurrence of 0.4 at 3 years) to 28% (similar effect sizes of photodynamic therapy are reported in both intermediate and high risk groups), this will be equivalent to a relative reduction of 30%.

Recruitment of 533 participants (214 recurrences) will detect an HR of 0.64 between experimental and control strategies and provide, using the log-rank test, 90% power at a two-sided 5% significance level. This calculation assumes 2.5 years incremental recruitment, a minimum of 3-year follow-up and a 6.4% follow-up attrition at end of year three. To achieve this, we plan to use 30 secondary care sites that would see new bladder cancers diagnoses, from which we will exclude patients with MIBC (20%) and, from the remaining NMIBCs, exclude low risk disease (50%). Furthermore, we predict only 30% of these patients will be recruited based on willingness to participate or missed opportunities for recruitment.

## Health economics analysis

A within-trial cost-effectiveness analysis will be conducted to calculate incremental cost per recurrence avoided and incremental cost per QALY gained over 3 years. Data on costs, recurrence and QALYs for each participant will be recorded in the trial and used to estimate mean cost, recurrence and QALYs for each intervention group. As the time horizon of the trial is 3 years, these data will be discounted at 3.5%.[16] The cost, recurrence avoided and QALY data will then be used to estimate incremental costs, recurrence avoided and QALYs and incremental cost per recurrence avoided and incremental costs per QALY.

An economic model will be developed to estimate relative rates of cost-effectiveness and cost–utility, at 3 years (to mirror the within trial analysis) and over a patient lifetime time horizon. An perspective will be taken for the cost calculations. The model takes the form of a Markov state transition model that describes the consequences of different diagnosis and treatment strategies in terms of clinical and cost outcomes.[6] The rates of recurrence and progression recorded with the 3-year follow-up of the trial will be used to model short-term recurrence and progression rates. Further data required for the model relates to the transition and other probabilities of events beyond the 3-year follow-up, including the risk of recurrence and progression, probabilities of receiving different types of intervention should progression or recurrences occur and risks of mortality (both from bladder cancer and other causes), will be sought through a structured systematic review of long-term outcomes of treatments of bladder cancer. The model will be used to produce estimates of costs, QALYs, recurrence rates and survival. Both costs and outcomes will be discounted at 3.5% in the base case analyses. Cost-effectiveness will be reported as incremental cost per QALY gained and incremental cost per recurrence avoided (at both 3 years and over the patient's lifetime). These data will be presented as point estimates and bootstrapping techniques will be used to estimate the statistical imprecision surrounding them. The results of this stochastic analysis will be presented as cost and QALY plots and as cost-effectiveness acceptability curves. Further deterministic sensitivity analyses will be conducted to explore other forms of uncertainty, for example, surrounding the choice of discount rate or around the unit costs of equipment. The model will be probabilistic, and distributions will be attached to all parameters, and the shape and type of distribution will depend on the data available and recommendations for good practice in modelling.[17]

## Patient and public involvement

Patient involvement was ensured at the early stages of protocol development and contributed to user-lead development of outcomes of value to patients in the design of the trial. Additionally, the patient journey of patient representatives was investigated through the diagnosis and treatment of bladder cancer, which includes an anonymised account impact on his quality of life. This helped understand the burden of the intervention on patients. A patient representative was involved as a coinvestigator and member of the trial steering committee helping manage and analyse the implications of the research.

## PHOTO Translational (PHOTO-T) side study

PHOTO-T aims to establish a well-characterised trial associated biorepository of longitudinal serially collected tissue samples (blood, urine and FFPE). The collection of samples from PHOTO patients is optional with every PHOTO-T consented participant collecting a urine and blood sample at baseline (pre-treatment/TURBT) and

3, 12, 24 and 36 months treatment follow-up or at recurrence (whichever comes first, predicted at 70% for the highest risk group over a trial period of 3 years). A FFPE core at baseline (plus recurrence, if occurs) will also be collected (online supplementary).

## DISCUSSION

Bladder cancer is the most frequent urothelial cancer, and the overall costs for treatment and follow-up remain higher than most other cancers.[18] Achieving complete resection of NMIBC with TURBT is associated with lower recurrence rates in follow-up. However, it is unclear if this translates into lower progression rates in long-term follow-up. PDD-guided initial TURBT is a technology that could improve resection and ultimately reduce recurrence and the need for further treatments.

Studies on PDD have demonstrated the efficacy of the technology using strict study entry requirements, for which translation into daily clinical practice is limited. Therefore, in the PHOTO trial, the effectiveness of the technology as part of routine care will be demonstrated with a pragmatic clinical trial design.

PHOTO trial includes measurement of HRQoL using EQ-5D at the time of initial treatment and surveillance. The measurement of HRQoL around the time of the cystoscopy and TURBT can be particularly dynamic due to an acute deterioration in health score associated with the invasive procedure followed by a typical rapid recovery.[11 19] Therefore, a side study was developed, where patients are recruited from the PHOTO trial to evaluate the acute deterioration in quality of life by suspected diagnosis or TURBT around the time of resection. This side study will use a time trade off exercise, and the outcomes will supplement the calculation of QALYs in the health economic model.

The high costs of bladder cancer to healthcare systems has usually been obtained from weak data, and the true costs are unclear. The pragmatic design of the PHOTO trial alongside the robust data collection for a full health economic evaluation will provide high-quality evidence of the burden of NMIBC for the NHS. Moreover, it will also provide a cost-effectiveness comparison of white light versus PDD-guided initial TURBT resections.

Evidence on the required cases for PDD-naïve surgeons to gain competency the technology is weak. This could act as a potential confounder on the clinical outcomes measured and therefore will be accounted for during analysis. Moreover, an evaluation of the learning curve of PDD will also be carried out using the forms filled in by surgeons.

The primary outcome of the study is time to recurrence measured from the day of randomisation to the day of subsequent biopsy with pathologically proven recurrence. If decrease in time to recurrence is associated with long-term patient benefits, the cost-effectiveness evaluation will provide further evidence for the NHS to decide on full adoption of the technology.

**Author affiliations**
[1]Urology, Northern Institute for Cancer Research, Newcastle upon Tyne, UK
[2]Urology and Head and Neck Trials Team, Institute of Cancer Research, London, UK
[3]Centre for Healthcare Randomised Trials (CHaRT), University of Aberdeen, Aberdeen, UK
[4]Urology and Head and Neck Trials Team, Institute of Cancer Research, London, UK
[5]Health Economics Group, Institute of Health and Society, Newcastle University, Newcastle, UK
[6]Health Economics Group, Institute of Health and Society, Newcastle University, Newcastle upon Tyne, UK
[7]Department of Urology, University College London Hospitals NHS Foundation Trust, London, UK
[8]Institute of Cellular Medicine, Newcastle University, Newcastle upon Tyne, UK
[9]Department of Surgery, University of Aberdeen, Aberdeen, UK
[10]Health Service Research Unit, University of Aberdeen, Aberdeen, UK
[11]Urology, Hampshire Hospitals NHS Foundation Trust, Winchester, UK
[12]Department of Urology, Western General Hospital, Edinburgh, UK
[13]Department of Medicine, University of Dundee, Dundee, UK
[14]Urology, South Tees Hospitals NHS Foundation Trust, Middlesbrough, UK
[15]Urology, Royal Liverpool and Broadgreen University Hospitals NHS Trust, Liverpool, UK
[16]Department of Urology, Royal Devon and Exeter NHS Foundation Trust, Exeter, UK
[17]Trial Management Group, PHOTO Trial, UK
[18]Urology, Northern Institute for Cancer Research, Newcastle upon Tyne, UK
[19]Edinburgh Clinical Trials Unit, University of Edinburgh, Edinburgh, UK
[20]Urology and Head and Neck Trials Team, The Institute of Cancer Research, London, UK

**Contributors** ZT and RH: conception and design of the work, drafting of the article, critical revision of the article and final approval of the version to be published. RL, AD and AM: conception and design of the work, data collection, drafting of the article, critical revision of the article and final approval of the version to be published. LV, JN and EH: conception and design of the work, critical revision of the article and final approval of the version to be published. SP: data collection, critical revision of the article and final approval of the version to be published. JS and GM: design of the work, critical revision of the article and final approval of the version to be published. PHOTO TMG members: trial management, data collection and final approval of the version to be published. Members of this group include: JDK, RP, JND, CR, HM, PM, GN, JC, HL, JM, ET and EC.

**Funding** This project was funded by the National Institute for Health Research HTA (project number 11/142/02). The PHOTO trial is sponsored by Newcastle upon Tyne Hospitals National Health Service Foundation Trust.

**Disclaimer** The views expressed are those of the author(s) and not necessarily those of the NHS, the NIHR or the Department of Health. Management of the study is divided between the Clinical Trials & Statistics Unit at the Institute of Cancer Research and Centre for Healthcare Randomised Trials at the University of Aberdeen. Independent trial steering committee members: Mr Peter Whelan (Chair), Dr Andrew Vickers, Professor Per-Uno Malmstrom, Mr Allen Knight and Dr Dan Sjoberg. Independent data monitoring committee members: Dr Angela Casbard (chair), Professor Diane Witham, Mr Robert Mills, Dr Ed Wilson and Mr Paul Silcocks.

**Competing interests** None declared.

**Patient consent for publication** Not required.

**Ethics approval** Favourable ethical opinion for this research was provided by the Newcastle & North Tyneside 2 ethics committee (REC reference number: 14/NE/1062) in July 2014.

**Provenance and peer review** Not commissioned; externally peer reviewed.

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
