## [Reviewer comments · BMJ Open]

ARTICLE DETAILS

TITLE (PROVISIONAL)	PHOTodynamic versus white light-guided treatment of non-muscle invasive bladder cancer: A study protocol for a randomised trial of clinical and cost effectiveness
AUTHORS	tandogdu, zafer; Lewis, Rebecca; Duncan, Anne; McDonald, Alison; Vale, Luke; Penegar, Steven; Shen, Jing; Kelly, John; Pickard, Robert; NDow, James; Ramsay, Craig; Mostafid, Hugh; Mariappan, Paramanathan; Nabi, Ghulam; creswell, Joanne; Lazarowicz, Henry; McGrath, John; Taylor, Ernest; Clark, Emma; MacLennan, Graeme; Norrie, John; Hall, Emma; Heer, Rakesh

VERSION 1 – REVIEW

REVIEWER	Catherine Poh Professor, University of British Columbia, Canada
REVIEW RETURNED	15-Oct-2018

GENERAL COMMENTS	This trial was open to recruitment on Oct 28, 2014 and has completed recruitment on February 14, 2018. (https://www.cancerresearchuk.org/about-cancer/find-a-clinical-trial/a-trial-white-light-blue-light-surgery-people-bladder-cancer-photo). The authors should add the following information to the manuscript: 1. The trial starting date and end date are not in the manuscript. The only date in the manuscript is the IRB approved date.2. How many centres participated in the study, clinicians' level of training (for example how many will be newly trained clinicians using PDD approach) and the reference for the estimated patients of 4590 in sections 2.9.3. The authors should put eligibility check list for patient screening as an appendix to the manuscript.4. There is no figure 2 in the manuscript.5. Figure 1 is not necessary. However, a picture to show the difference between WL and PDD assessment will be very appealing on the visual enhancement using PDD.6. In the discussion the authors should expand more on calibration of the clinicians/surgeons in the eligibility assessment and the treatment using PDD are not described in the protocol. Although it was mentioned in the discussion as 'Evidence suggests that 20 cases are required for PDD naïve surgeons to gain competency the technology. This could act as a potential confounder on the clinical outcomes measured and therefore will be accounted for during analysis.' These sentences did not explain the number of new clinicians requiring the 20 cases to develop competency in management and how to calibrate their success. Additionally, if many new sites, this might reduce the number of 'effective cases' in the PDD group.
--

	Since the study has completed its patient recruitment, the change of protocol is not possible. However the following considerations might help the authors during data analysis.  1. Economic analysis. The author should also consider patients' out-of-pocket cost, e.g., travel and loss of work for care takers etc. 2. In section 2.9 - How the sample size calculation (n= 533) and estimated number of recurrence (n=214) are not clear? Based on the estimated recurrence rates in white light (40%) and PDD (28%) groups, the estimated recurrence should be 182. The author should also calculate 10% loss during the follow-up to make sure the trial has enough recruitment. This was not included in sample size justification. 3. Ta and T1 are more similar disease than Tis. Will these make difference in visualization, hence outcome of recurrence with the help of PDD? I would suggest to perform minimization during the randomization (too late now) or subgroup analysis if the trial has enough number of the Tis group.
--	---

VERSION 1 – AUTHOR RESPONSE

1. The trial starting date and end date are not in the manuscript. The only date in the manuscript is the IRB approved date.

These have been added thanks.

2. How many centres participated in the study, clinicians' level of training (for example how many will be newly trained clinicians using PDD approach) and the reference for the estimated patients of 4590 in sections 2.9.

22 centers contributed to the study. This statement has been added to the manuscript. The estimate of 4590 has been removed to avoid any further confusion.

3. The authors should put eligibility check list for patient screening as an appendix to the manuscript.

The eligibility criteria are listed in the manuscript under section 2.9 Participants. We thought that adding it as a checklist could cause replication. Thanks.

4. There is no figure 2 in the manuscript.

We apologise for the error and have uploaded figure 2 that shows the study design. Thanks for highlighting this.

5. Figure 1 is not necessary. However, a picture to show the difference between WL and PDD assessment will be very appealing on the visual enhancement using PDD.

Thanks for this comment. We have obtained images of a patient with both WL and PDD that correspond to an area with CIS. This, should provide a good example for readers.

6. In the discussion the authors should expand more on calibration of the clinicians/surgeons in the

eligibility assessment and the treatment using PDD are not described in the protocol. Although it was mentioned in the discussion as 'Evidence suggests that 20 cases are required for PDD naïve surgeons to gain competency the technology. This could act as a potential confounder on the clinical outcomes measured and therefore will be accounted for during analysis.' These sentences did not explain the number of new clinicians requiring the 20 cases to develop competency in management and how to calibrate their success. Additionally, if many new sites, this might reduce the number of 'effective cases' in the PDD group.

Thanks for this comment. To overcome any confusion, we have re-phrased the discussion section as follows: "Evidence on the required cases for PDD naïve surgeons to gain competency the technology is weak. This could act as a potential confounder on the clinical outcomes measured and therefore will be accounted for during analysis. Moreover, an evaluation of the learning curve of PDD will also be carried out using the forms filled in by surgeons."

We also collect data on the previous experiences of surgeons, and this will be accounted for in analysis as a potential confounder. This explanation is provided in the section 2.6.3 as follows: "The effect of PDD guided resection experience (learning curve) on clinical effectiveness: A subgroup analysis comparing outcomes from PDD-experienced and PDD-naïve surgeons (determined at baseline) will be conducted. Also for PDD-naïve surgeons an assessment of learning curve will be undertaken by comparing increasing experience and recurrence, in both PDD and WL resections."

Additionally in section 2.8 the following explanation is available: "Effect of PDD guided resection experience on clinical effectiveness: All recruiting surgeons will complete a learning curve questionnaire to elicit their white light and PDD resection experience prior to any recruitment. The subsequent accruing experience of each surgeon will be captured on case report forms. Early recurrence (12 weeks) will be used as a proxy of incomplete resection.

We thank for the reviewer highlighting this issue and hope to avoid any confusion for future readers by the changes made.

Since the study has completed its patient recruitment, the change of protocol is not possible. However the following considerations might help the authors during data analysis.

1. Economic analysis. The author should also consider patients' out-of-pocket cost, e.g., travel and loss of work for care takers etc.

We send out a postal Costs Questionnaire' at 30 months post randomisation which collects patients out of pocket costs at this time point (as well as the health utilisation collected HE outcomes collected at 3, 6, 12, 24 & 36 months f/up). In the protocol it is stated under section 12.2.1: *Costs for healthcare services will be obtained from standard sources such as NHS reference Healthcare Resource Group (HRG) tariffs and the British National Formulary, from relevant manufacturers and suppliers and directly from secondary care centres. For each participant, measures of resource-use will be combined with unit costs to provide cost for that participant.*

2. In section 2.9 - How the sample size calculation (n= 533) and estimated number of recurrence (n=214) are not clear? Based on the estimated recurrence rates in white light (40%) and PDD (28%) groups, the estimated recurrence should be 182. The author should also calculate 10% loss during the follow-up to make sure the trial has enough recruitment. This was not included in sample size justification.

We would like to thank the reviewer for the feedback. Inflation for loss to follow-up was based on the BOXIT trial as 0.56% after year 1, 3.1% after year 2 and 6.4% after year 3. The BOXIT trial was conducted by similar clinical units and managed by the same clinical trials unit.

The sample size calculation is based on a log-rank test with 90% power, 2-sided 5% significance and assumed 40% recurrence rate (i.e. 60% recurrence-free) at 3 years in intermediate risk group. The trial seeks to detect a hazard ratio of 0.64 which translates to a 30% relative reduction in recurrences at 3 years from 40% to 28% i.e. an increase in the proportion of patients recurrence-free from 60% to 72% requiring 214 events.

The target sample size of 533 patients was based on a 2.5 years recruitment period (with recruitment weights of 0.6 0.13 0.21 0.29 and 0.31 for each 6 month period) and had a minimum of 3 years follow-up on all patients.

3. Ta and T1 are more similar disease than Tis. Will these make difference in visualization, hence outcome of recurrence with the help of PDD? I would suggest to perform minimization during the randomization (too late now) or subgroup analysis if the trial has enough number of the Tis group. Thanks for highlighting this aspect of the trial. We are planning to conduct ad-hoc analysis if group numbers allow as such. In addition, we have stratified cases according to risk groups based on visual and imaging findings. Flat and velvety changes that indicate CIS (Tis) are also part of the inclusion criteria.